# Epidemiology of Simultaneous Medullary and Papillary Thyroid Carcinomas (MTC/PTC): An Italian Multicenter Study

**DOI:** 10.3390/cancers11101516

**Published:** 2019-10-09

**Authors:** Marialuisa Appetecchia, Rosa Lauretta, Agnese Barnabei, Letizia Pieruzzi, Irene Terrenato, Elisabetta Cavedon, Caterina Mian, Maria Grazia Castagna, Rossella Elisei

**Affiliations:** 1Oncological Endocrinology Unit, IRCCS Regina Elena National Cancer Institute, 00144 Rome, Italy; rosalauretta@yahoo.it (R.L.); agnese.barnabei@ifo.gov.it (A.B.); 2Endocrinology Unit, University Hospital of Pisa, 56121 Pisa, Italy; pieruzziletizia@gmail.com (L.P.); rossella.elisei@med.unipi.it (R.E.); 3Biostatistics-Scientific Direction, IRCSS Regina Elena National Institute, 00144 Rome, Italy; irene.terrenato@ifo.gov.it; 4Istituto Oncologico Veneto, University of Padua, 35100 Padua, Italy; elisabetta.cavedon@unipd.it (E.C.); caterina.mian@unipd.it (C.M.); 5Endocrinology Unit, Policlinico S.M. alle Scotte, 53100 Siena, Italy; mariagraziacastagna@hotmail.com

**Keywords:** medullary thyroid cancer, papillary thyroid cancer, epidemiology

## Abstract

*Background*: The concomitant presence of papillary thyroid cancer (PTC) and medullary TC (MTC) is rare. In this multicentric study, we documented the epidemiological characteristics, disease conditions and clinical outcome of patients with simultaneous MTC/PTC. *Methods:* We collected data of patients with concomitant MTC/PTC at 14 Italian referral centers. *Results:* In total, 183 patients were enrolled. Diagnosis was mostly based on cytological examination (*n* = 58, 32%). At diagnosis, in the majority of cases, both PTC (*n* = 142, 78%) and MTC (*n* = 100, 54%) were at stage I. However, more cases of stage II–IV were reported with MTC (stage IV: *n* = 27, 15%) compared with PTC (*n* = 9, 5%). Information on survival was available for 165 patients: 109 patients (66%) were disease-free for both PTC and MTC at the last follow-up. Six patients died from MTC. Median time to progression was 123 months (95% confidence interval (CI): 89.3–156.7 months). Overall, 45% of patients were disease-free after >10 years from diagnosis (125 months); this figure was 72.5% for PTC and 51.1% for MTC. *Conclusions:* When MTC and PTC are concurrent, the priority should be given to the management of MTC since this entity appears associated with the most severe impact on prognosis.

## 1. Introduction

Papillary thyroid carcinoma (PTC) accounts for the wide majority (approximately 85%) of thyroid cancers [1]. The incidence of this disease has largely increased over the last decades, primarily due to advances in diagnostic approaches [2]. According to current knowledge, PTC is an umbrella term that encompasses several tumor types with mutually exclusive mutations, in most cases *BRAF* V600E, followed by *RAS* (15%) and chromosomal rearrangements leading to the expression of the kinase domains of BRAF or of receptor tyrosine kinases, such as RET, NTRK, and ALK (12%) [1]. Different mutations result in different disease behavior. However, most PTCs are clinically indolent, consistent with their simple genome characterized by few copy number alterations and a low mutational density [1,3]. 

Medullary thyroid carcinoma (MTC) is another type of thyroid carcinoma. It is much rarer than PTC, accounting for 3–5% of all thyroid cancers [1]. In three out of four patients, MTC is sporadic; less often, it represents the dominant component of the hereditary multiple endocrine neoplasia (MEN) type 2 syndromes, MEN2A and MEN2B. *RET* is the driver oncogene in MTC, followed by *RAS* mutations and RET or ALK fusions [4,5]. The clinical aggressiveness of MTC is related to *RET* mutation. When shared oncogenes between the two malignancies were sought, no common genetic alterations were found [6].

The concomitant presence of PTC and MTC is a rare event, described in the literature mainly in anecdotal reports [7,8,9,10,11,12,13,14,15,16,17,18,19,20,21,22,23,24,25,26,27], and in a few studies [6,28,29,30,31,32]. It is debated in the literature whether the concomitant presence in the same subject of PTC and MTC is random or whether it depends on a common gene alteration. More importantly, the clinical outcomes of patients with concurrent PTC and MTC require further investigation in a large sample of patients. In this multicenter study, we documented the epidemiological characteristics, disease conditions and clinical outcome of patients with simultaneous MTC/PTC.

## 2. Patients and Methods

### 2.1. Study Setting and Design

We collected data of patients with concomitant MTC/PTC diagnosed between 1992 and 2014 at 14 Italian referral centers located all over the country. Local Ethical Committees—Comitato Etico Centrale IRCCS Lazio Sez. IRCCS IFO-Fondazione G. B. Bietti—approved the study design on 12 July 2016 (approval code: RU/8684; ethic code: RS 827/16) and all patients had signed an informed consent to the use of their personal data for research purposes. 

### 2.2. Patients and Procedures

Clinical charts of patients treated in the participating Centers from 1992 to 2014 were reviewed to identify those with concomitant MTC/PTC (foci had to be distinct in all cases). No other inclusion/exclusion criteria were applied. All patients were diagnosed and managed according to the standard practice of each center where they were followed.

For each patient with concomitant MTC/PTC, we reviewed demographic and clinical data (blood tests and imaging results), epidemiological characteristics, pathological conditions and clinical outcomes. For the staging of both PTC and MTC, the tumor, node and metastases (TNM 7th edition) staging system was applied. 

Somatic and germline *RET* gene mutation data were collected, when possible (Sanger sequencing). Analysis was performed on both tissues and blood samples.

### 2.3. Data Analysis

We explored patients’ and disease features at baseline, and between the same characteristics and clinical outcomes, in terms of metastatic status and progression-free survival (PFS; defined as the time from diagnosis to documented progression according to the RECIST criteria or death, whichever occurred first). Descriptive statistics were computed for all the variables of interest. PFS was evaluated according to the Kaplan–Meier product-limit method. Stratified analysis by specific demographic and pathological characteristics were also conducted, overall and for both PTC and MTC. Associations between variables were evaluated by Pearson’s Chi-Square test. *p* < 0.05 was considered statistically significant. All the statistical analyses were carried out using SPSS software (SPSS version 21.0, IBM, Armonk, NY, USA).

## 3. Results

### 3.1. Patients 

In total, 183 patients were enrolled (mean age: 56 ± 13 years; range: 16–84 years; 39 (21%) aged ≤45 years; 105 (58%) females). Table 1 depicts their baseline characteristics, including laboratory examinations, respectively; Appendix A lists oncological comorbidities. Suspicious diagnosis was mostly based on cytological examination (*n* = 58, 32%; examination was decided due to medullary carcinoma in 20 cases and to papillary carcinoma in 38), followed by cytological examination + basal calcitonin (BCT) and BCT only (*n* = 39 each; 21%). One-third of patients who underwent surgical procedure were submitted to total thyroidectomy with a central lymph node dissection (33%). Overall, 44% of patients (*n* = 81) were positive for pre-surgery anti-thyroglobulin antibodies and 14% (*n* = 25) for anti-thyroperoxidase antibodies (AbTPO). Diagnosis was then confirmed by histological examination in all cases.

### 3.2. Anatomo-Pathological Features Analysis

Table 2 shows the pathological features of our series. At diagnosis, in the majority of cases, both PTC (*n* = 142, 78%) and MTC (*n* = 100, 54%) were at stage I of the disease (Table 2); however, more cases of stage II–IV were reported with MTC (stage IV: *n* = 27, 15%, compared with nine cases, 5% for PTC). Median follow-up from diagnosis was 32 months (range: 0–261). 

*RET* mutation was reported in 24 patients out of 112 in whom it was evaluated (21%). The mutation was germinal in 5/24 subjects (21%) and somatic in the remaining 19 subjects. Appendix A displays the specific mutations in this gene: V804M was the most frequently reported (*n* = 8, 33%).

### 3.3. Clinical Outcomes

Overall, 18 (9.8%) patients were excluded from PFS analysis because they were lost to follow-up or it was not possible to give a final evaluation in terms of presence of disease due to lack of specific information for one of the two tumors. Therefore, 165 patients had available information about their overall survival (OS) and PFS status. Of these, 109 patients (60%) were disease-free both for PTC and MTC at the last follow-up (Table 3). In the remaining 40% of cases, persistent/recurrent disease was reported (biochemical disease, loco-regional disease or distant metastases). Six patients died from MTC (one MTC stage IVa and five MTC stage IVc), and another patient died from pancreatic cancer; therefore, at the last follow-up, 158 (94%) patients were alive.

The Kaplan–Meier method gave an estimated median time to progression equal to 123 months (95% confidence interval (CI): 89.3–156.7 months) (Figure 1A). Overall, 45% of patients were disease-free after more than 10 years from diagnosis (125 months); this figure was 72.5% for PTC and 51.1% for MTC (Figure 1B,C). 

### 3.4. Kaplan–Meier Stratified Analyses

No differences were observed in conducting stratified analysis except for stage for specific MTC analysis (Appendix A, stage I + II median time to progression 175 months (95% CI: 103.5–246.5) vs. stage III + IV 74 months (95% CI: 14.6–133.4; *p* < 0.0001). Analysis of survival by tumor size is reported in Appendix A.

## 4. Discussion

The concomitant presence of PTC and MTC is a very rare event in clinical practice, and debate remains open as to whether this event should be considered coincidental or rather the result of a common genetic alteration [6]. Beside this—and possibly with greater relevance for clinical practice—the clinical outcomes of patients with concomitant PTC and MTC have only been poorly investigated, and only small series of patients were analyzed given the rarity of this occurrence [6,28,29,30,31,32]. 

In this Italian multicentric study, we investigated the epidemiological characteristics, disease conditions and clinical outcome in a large series of 183 patients with simultaneous MTC/PTC, and we were able to provide a picture of their characteristics. Remarkably, most patients were aged ≥45 years, in line with a previous study, which suggested that patients with concomitant MTC/PTC are older than those with MTC alone [29] To our knowledge, this is one of the largest series, if not the largest, of subjects with concomitant MTC/PTC observed to date. Moreover, given the length of the follow-up (median follow-up, 32 months from diagnosis, with a maximum of 261 months, i.e. approximately 22 years), we were able to document the clinical outcomes of this population. However, we must acknowledge that the retrospective nature of our analysis hampers the results, especially due to the high number of missing data.

Although with these limitations, it appears that the prognosis of patients with concomitant MTC and PTC is good, with a 97% survival rate and a 66% of patients being progression-free at the last follow-up; 45% of patients were disease-free at more than 10 years from diagnosis. This finding is in line with a previous study, which show the relatively good prognosis of patients with simultaneous MTC/DTC compared to MTC only (10-year survival rates 87% vs. 81%) [33]. Interestingly, the survival rates observed in our patient cohort were found to be lower compared to those reported in stage I sporadic MTC patients (i.e., 100% at 10 years) [34]. Moreover, despite the pre-operative calcitonin levels were less than 10 pg/mL (levels associated with 97% survival rates at 10 years), only 45% were disease-free. Thus, the coexistence of both MTC and PTC might confer a worse prognosis compared to MTC only patients. Nevertheless, this hypothesis would require further investigation in a larger and more diverse population comprising MTC/PTC, MTC-only and PTC-only patients. Based on our preliminary observations, the prognosis appears driven by MTC. Out of the seven patients who died over the follow-up period, six MTC-related deaths were reported, all in patients with stage IV disease. Moreover, the Kaplan–Meier analysis showed that the overall progression pattern closely mirrors the pattern reported for MTC only. One can argue that this finding could be somehow expected, since MTC is a much more aggressive tumor than PTC. However, if MTC is early diagnosed, when it is still localized [35], we can expect to achieve surgical cure. From our data, the diagnosis of MTC was often performed too late, when the tumor was already not curable anymore. In our series, not all cases of MTC were diagnosed by measurement of serum calcitonin, that has been demonstrated to be more sensitive and specific than cytology [36]. Remarkably, as in the above-mentioned study series [6,28,29,30,31,32], in our series, 48 cases for sure, and likely also others among those for whom we missed the data, were submitted to an inadequate surgical treatment for MTC (i.e., total thyroidectomy alone without at least central compartment lymphadenectomy), which in many circumstances jeopardizes the cure of the patients.

RET is an oncogene involved in the development of sporadic and hereditary MTC while chromosomal translocations activating RET occur in 20–30% of patients with PTC. In our patient cohort, only 24 patients were harboring RET mutations and most of them, independently from the predominant histotype, were disease-free or with persistent disease. Our observations contrast with previous studies reporting that RET+ MTC were mostly associated with a worse prognosis [37]. Moreover, the RET mutation most frequently associated with the most aggressive behavior of the MTC, namely M918T, was found in only two patients.

In our analysis, we failed to identify any characteristic able to distinguish between MTC and PTC or to predict outcome, with the exception of MTC stage at diagnosis, with a more advanced stage being associated with higher risk of progression. Our finding seems consistent with the predictive relevance of stage at diagnosis reported in a small cohort of 19 patients with concurrent MTC/PTC [38]. Among the patients with recurrent disease, the subtype of recurrence was consistent with the preoperative diagnosis and with the subtype with the more advanced stage at the time of initial surgery. However, it should be noted that, in our patient population, most PTCs were at stage I and of classic and follicular variants, which are reported historically being low-risk subtypes [39]. Another relevant finding is the role of serum calcitonin evaluation, which helped us in the diagnosis of about 40% of cases. 

## 5. Conclusions

Our study shows that, when MTC and PTC are concurrent in the same patient, the priority should be given to the management of MTC—however, according to the specific characteristics of the two tumors—since this entity appears to be the one with the most severe impact on prognosis. Since patients with concomitant MTC and PTC have one or more thyroid nodules, in their presurgical work up, calcitonin should be always included, even in the presence of an already positive cytology for PTC. Moreover, our findings further support the notion that the early diagnosis of MTC is still an unmet need both when it is isolated and when associated to another thyroid cancer such as PTC [38]. Further studies aimed at investigating the molecular signature of concomitant MTC and PTC may hold promise to pave the way for the identification of candidate target genes for therapy. Gene-expression sequencing and microarrays along with genome-wide association studies may aid in the identification of cancer-specific germline and somatic mutations, which can contribute to more sensitive diversification of cancer subtypes and facilitate early diagnosis. Collectively, genomics-driven approaches are expected to lead to a more personalized treatment for patients with simultaneous MTC/PTC. 

## Figures and Tables

**Figure 1 cancers-11-01516-f001:**
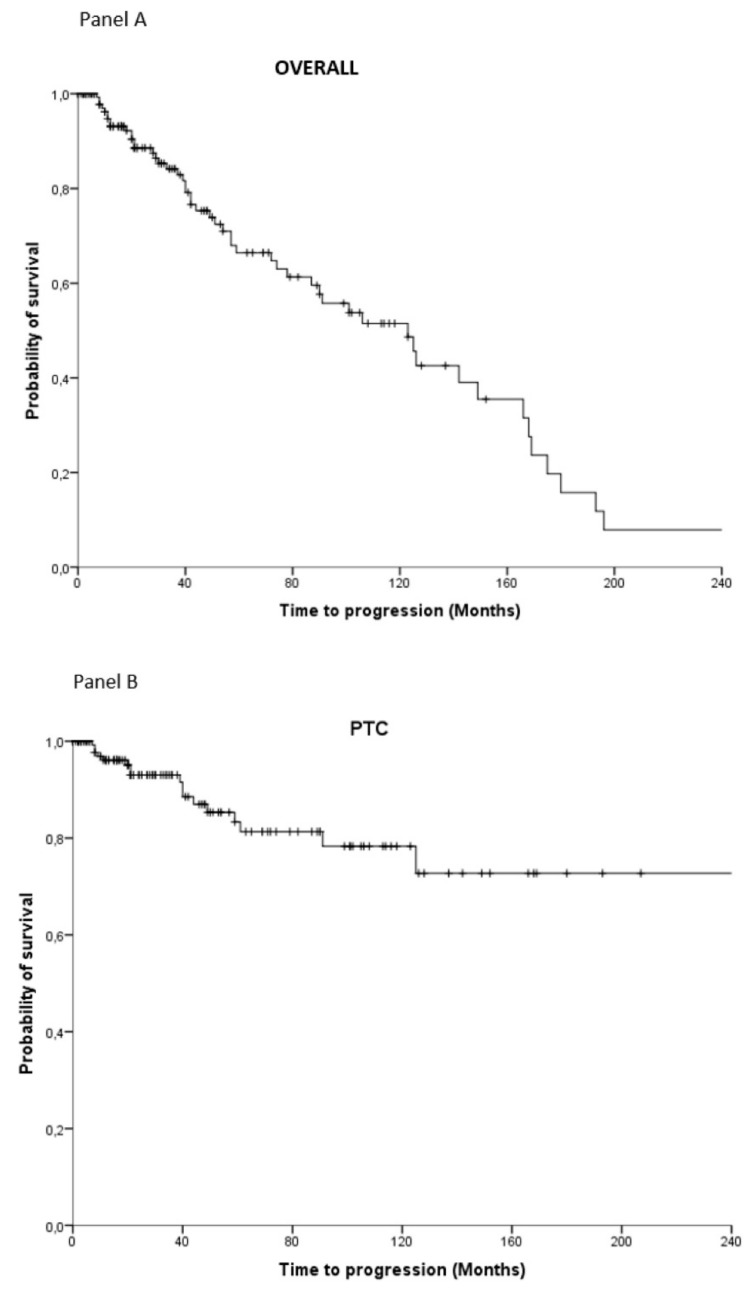
Overall progression-free survival (**A**) and specific progression-free survival for PCT (**B**) and MTC (**C**) only.

**Table 1 cancers-11-01516-t001:** Baseline characteristics of the 183 patients.

N° of Cases	*N*	%
	183	
Observed follow-up period (min–max); months	30 (0–261)	
Age at diagnosis: (years):		
– Mean (SD)	56.2 (13.4)	
– ≤45 years	39	20
– >45 years	143	79
– Unknown	1	1
Gender:		
– Female*s*	105	58
– Males	78	42
Circumstances of diagnosis:		
– Fine needle aspiration cytology	58	32
– Fine needle aspiration cytology + basal calcitonin	39	21
– Basal calcitonin	39	21
– Incidental	19	10
– Family history	5	3
– Family history + basal calcitonin	2	1
– Unknown	21	12
Presence of non-oncologic comorbidity:	*N*	%
– None	13	7
– Osteoporosis/osteopenia	11	6
– Surrenal/hypophysis adenomas	5	3
– Other	114	62
– Unknown	40	22
Presence of oncologic comorbidity:		
– Yes	21	12
– No	114	62
– Unknown	40	22
Familiar thyroid diseases:		
– Yes	33	18
– No	95	52
– Unknown	55	30
Oncologic family history:		
– Yes	38	21
– No	77	42
– Unknown	68	37
Familiar thyroid cancer:		
– Yes	18	10
– No	165	90
Thyroid goiter		
– Yes	98	54
– No	61	33
– Unknown	24	13
FT4 (free thyroxine):		
– EU (euthyroidism)	81	44
– EU in treatment	4	2
– Hypothyroidism	3	2
– Hyperthyroidism	13	7
– Unknown	82	45
Pre-surgical calcitonin (ng/L), mean (SD)	699.2 (1557.0)	
Pre-surgical anti-thyroglobulin antibodies		
– Negative	102	56
– Positive	32	17
– Unknown	49	27
Pre-surgical anti-thyroid peroxidase antibodies:		
– Negative	109	59
– Positive	25	14
– Unknown	49	27

**Table 2 cancers-11-01516-t002:** Tumor characteristics (*N* = 183).

Tumor Characteristics	*N*	%
Histology:		
– PTC classic variant + MTC	77	42
– PTC follicular variant + MTC	51	28
– PTC (other) + MTC	55	30
Staging* of PTC:		
– 1	142	78
– 2	8	4
– 3	10	5
– 4	9	5
– Unknown	14	8
Staging* of MTC:		
– 1	100	54
– 2	11	6
– 3	27	15
– 4	27	15
– Unknown	18	10
PTC:		
– Tx	2	1
– T1	152	83
– T2	10	6
– T3	16	9
– T4	3	1
– Nx	24	13
– N0	136	74
– N1–2	23	13
– Mx	48	26
– M0	134	73
– M1	1	1
MTC:		
– Tx	11	6
– T1	113	62
– T2	22	12
– T3	32	18
– T4	5	2
– Nx	25	14
– N0	106	58
– N1–2	52	28
– Mx	49	27
– M0	123	67
– M1	11	6
PTC size:		
– ≤1 cm	148	81
– >1 cm	29	16
– Unknown	6	3
MTC size:		
– ≤1 cm	86	47
– >1 cm	85	46
– Unknown	12	7
RET mutation:		
– Yes	24	13
– No	88	48
– Unknown	71	39

*Anatomic stage according to the 7th Edition of AJCC staging system.

**Table 3 cancers-11-01516-t003:** Clinical outcomes.

Clinical Outcomes	N	%
**Overall cancer-specific survival outcome**		
Disease free	109	60
Biochemical disease	32	18
Distant metastasis:	10	5
– Locoregional disease	8	4
– Death by MTC	6	3
– Not evaluable	18	10
**PTC outcome**		
Disease free	143	78
Biochemical disease	17	9
Distant metastasis	1	1
Locoregional disease	1	1
Not evaluable	21	11
**MTC outcome**		
Disease free	119	65
Biochemical disease	24	13
Distant metastasis	10	5
Locoregional disease	7	4
Death by MTC	6	3
Not evaluable	17	10

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
