# Peer review of "Epidemiology of Simultaneous Medullary and Papillary Thyroid Carcinomas (MTC/PTC): An Italian Multicenter Study"

_cancers, 2019, doi:10.3390/cancers11101516_

Round 1
Reviewer 1 Report
The Authors replied to all my comments and applied changes have improved the manuscript. I have only some minor comments:
All abbreviations should be explained when used for the first time (line 102, abbreviation TX) Section 3.2 Anatomo-Pathological features analysis includes molecular data analysis as well so the title of this section should be changed.Reviewer 2 Report
Main comments have been modified, and the manuscript is more clear .
Only a remark about the stage : the international nomenclature for the stage is I, II, III, IV.The correct nomenclature needs to be changed in TABLE 2, and p 11 line 197 and 235
This manuscript is a resubmission of an earlier submission. The following is a list of the peer review reports and author responses from that submission.
Round 1
Reviewer 1 Report
Authors evaluated whether the concomitant presence in the same subject of PTC and MTC is randomly coincidental or whether it depends on a common gene alteration. They also evaluated the clinical outcomes of 183 patients with concurrent PTC and MTC in the largest sample size to date.
Major points
The major concern with the manuscript is the methods of pathologic diagnosis and clinicopathologic evaluation.
1) Diagnosis of PTC/MTC was done by cytology (n = 58), cytology with basal calcitonin (n = 39), basal calcitonin (n = 39), casual (n = 19), family history with basal calcitonin (n = 2), and unknown (n = 26). Dose “cytology” mean fine needle aspiration cytology? The definitive diagnosis of the thyroid tumor should be based on histopathologic examination of surgically resected specimens.
2) It is not clear whether the simultaneous tumors are synchronous or metachronous, or both.
3) Every table should be clear and concise and have adequate title. Tables are too complicated to understand. Additional information of pT, pN and M stage for PTC and MTC should be added in table 2.
4) How many cases of PTC and MTC were incidentally found?
5) It is recommended to do sub-analysis for clinical outcomes in patients presenting with clinically evident tumors when microcarcinomas for PTC and MTC are excluded from all cases.
6) I conclusion, this study suggests that when MTC and PTC are concurrent in the same patient the priority should be given to the management of MTC since this entity appears to be the one with the most severe impact on prognosis. However, this conclusion is not clear. Patients can also present with a small MTC coexisting with aggressive PTC with advanced stage.
7) Authors need to discuss their initial questions about whether simultaneous MTC and PTC arise coincidentally or not.
Minor points
1) The citation of (SPSS version 21.0, SPSS Inc., Chicago, IL, USA) is not correct. Versions that were produced by SPSS Inc. before the IBM acquisition (Versions 18 and earlier) would be given an origin or publisher of SPSS Inc. in Chicago. Versions that were released after the acquisition would be given an origin or publisher of IBM Corp. in Armonk, NY.
2) There is a typo “follicolar” in table 3.
Author Response
Reviewer #1
Authors evaluated whether the concomitant presence in the same subject of PTC and MTC is randomly coincidental or whether it depends on a common gene alteration. They also evaluated the clinical outcomes of 183 patients with concurrent PTC and MTC in the largest sample size to date.
Thanks for your interest in our manuscript.
Major points
The major concern with the manuscript is the methods of pathologic diagnosis and clinicopathologic evaluation.
1) Diagnosis of PTC/MTC was done by cytology (n = 58), cytology with basal calcitonin (n = 39), basal calcitonin (n = 39), casual (n = 19), family history with basal calcitonin (n = 2), and unknown (n = 26). Dose “cytology” mean fine needle aspiration cytology? The definitive diagnosis of the thyroid tumor should be based on histopathologic examination of surgically resected specimens.
We revised the text to better clarify the diagnostic approach. With “Cytology” we indicated patients who underwent a needle aspiration. We clarified this in the text.
2) It is not clear whether the simultaneous tumors are synchronous or metachronous, or both.
Due to the retrospective nature of the study, it is not possible to go back to which of the two tumors was found before. We can only affirm that they were surely synchronous at the time of histological response.
3) Every table should be clear and concise and have adequate title. Tables are too complicated to understand. Additional information of pT, pN and M stage for PTC and MTC should be added in table 2.
In order to simplify the tables and give more readable results, we decided to report only the disease stage in Table 3 and not to add any other variables concerning the disease status. Furthermore, we made tables clearer also by a graphical point of view. Information on TNM was added in a supplementary table.
4) How many cases of PTC and MTC were incidentally found?
This information is now reported in Table 1.
5) It is recommended to do sub-analysis for clinical outcomes in patients presenting with clinically evident tumors when microcarcinomas for PTC and MTC are excluded from all cases.
Thanks for this major comment. We added, as supplementary material, the three PFS curves estimated by the Kaplan-Meier method considering only tumors greater than 1 cm, that are for the overall sample (both PTC and MTC greater than 1 cm), only for PTC and only for MTC.
6) I conclusion, this study suggests that when MTC and PTC are concurrent in the same patient the priority should be given to the management of MTC since this entity appears to be the one with the most severe impact on prognosis. However, this conclusion is not clear. Patients can also present with a small MTC coexisting with aggressive PTC with advanced stage.
We have reworded the text to better clarify our thoughts.
7) Authors need to discuss their initial questions about whether simultaneous MTC and PTC arise coincidentally or not
We commented on this in the text.
Minor points
1) The citation of (SPSS version 21.0, SPSS Inc., Chicago, IL, USA) is not correct. Versions that were produced by SPSS Inc. before the IBM acquisition (Versions 18 and earlier) would be given an origin or publisher of SPSS Inc. in Chicago. Versions that were released after the acquisition would be given an origin or publisher of IBM Corp. in Armonk, NY.
Reconciled.
2) There is a typo “follicolar” in table 3.
Fixed.
Reviewer 2 Report
Comments to Authors:
The Authors presented a large cohort of patients with simultaneous presence of MTC and PTC, giving an extensive epidemiological characteristics and clinical outcome of patients. The manuscript is well written, however few issued need additional comments and corrections:
1. The Authors write that somatic and germline RET mutation data were collected and there were 24 RET(+) cases out of 112 studied. Were all these 112 cases analyzed for somatic and germline mutations (analysis of tumor and blood samples was performed for all cases?)? Table S1 should contain information which of the presented mutations are somatic and which are germline.
2. Line 90- Authors write that in MOST cases patients were submitted to total thyroidectomy and central lymph node dissection, however, there is an information that only 33% of patients underwent this procedure. This should be corrected.
3. Twenty one cases exhibited oncologic comorbidity- these should be listed under the table.
4. Line 102- font needs correction.
5. Immunohistochamical analysis- I understand that in 77 cases there was classic PTC? When it comes to other PTC variants these should be listed.
6. How many of patients had distinct separate foci of MTC and PTC and how many presented mixed medullary and papillary carcinoma?
7. There should be an information how many patients from all centers analyzed had only MTC in order to show the % of simultaneous occurrence of MTC and PTC. The Authors could also analyze the age of these patients, since there are studies demonstrating that patients with PTC/MTC are older comparing to patients with MTC alone (Kim WG et al., Clin Endocrinol, 2010).
8. The results should be discussed more in relation to other studies. For example the observation that patients with concomitant MTC and DTC had good prognosis was previously described by Wong RL et al in 2012 (Ann Surg Oncol).
9. I would include the study of Beninato T et al from 2017 (World Journal of Endocrine Surgery) in the discussion, especially in the context of the preoperative diagnosis influence on the outcome.
10. The Authors write that they failed to identify any characteristic able to distinguish between MTC and PTC or to predict outcome, with the exception of MTC stage. However, there should be a comment that most PTCs were at stage 1 and of classic and follicular variants, which are less aggressive.
Author Response
Reviewer #2
The Authors presented a large cohort of patients with simultaneous presence of MTC and PTC, giving an extensive epidemiological characteristics and clinical outcome of patients. The manuscript is well written, however few issued need additional comments and corrections:
Thanks for your interest in our manuscript.
1. The Authors write that somatic and germline RET mutation data were collected and there were 24 RET(+) cases out of 112 studied. Were all these 112 cases analyzed for somatic and germline mutations (analysis of tumor and blood samples was performed for all cases?)? Table S1 should contain information which of the presented mutations are somatic and which are germline.
In Table 3 we reported the information about RET mutation. Twenty-four patients resulted RET positive, 88 were RET negative and for 71 patients we did not have any information. As we reported in the text, in 5/24 patients the mutation was germinal while in the remaining 19 patients we observed a somatic mutation. Analysis was performed both on tissues and blood samples.
2. Line 90- Authors write that in MOST cases patients were submitted to total thyroidectomy and central lymph node dissection, however, there is an information that only 33% of patients underwent this procedure. This should be corrected.
We reformulated the sentence to enhance clarity.
3. Twenty-one cases exhibited oncologic comorbidity- these should be listed under the table.
Available information was included in a Supplementary table.
4. Line 102- font needs correction.
Fixed.
5. Immunohistochamical analysis- I understand that in 77 cases there was classic PTC? When it comes to other PTC variants these should be listed.
We added a clearer information in table 3: the 77 cases labeled “PTC+MTC” were PTC classic variant.
6. How many of patients had distinct separate foci of MTC and PTC and how many presented mixed medullary and papillary carcinoma?
Foci were distinct in all cases. We clarified this in the text.
7. There should be an information how many patients from all centers analyzed had only MTC in order to show the % of simultaneous occurrence of MTC and PTC. The Authors could also analyze the age of these patients, since there are studies demonstrating that patients with PTC/MTC are older comparing to patients with MTC alone (Kim WG et al., Clin Endocrinol, 2010).
Unfortunately, we did not have a control group composed only by MTC cases. We did not plan to collect this information in the study design. We commented on the age of patients in the text and added the reference you suggested.
8. The results should be discussed more in relation to other studies. For example, the observation that patients with concomitant MTC and DTC had good prognosis was previously described by Wong RL et al in 2012 (Ann Surg Oncol).
We revised the Discussion accordingly.
9. I would include the study of Beninato T et al from 2017 (World Journal of Endocrine Surgery) in the discussion, especially in the context of the preoperative diagnosis influence on the outcome.
Included as requested.
10. The Authors write that they failed to identify any characteristic able to distinguish between MTC and PTC or to predict outcome, with the exception of MTC stage. However, there should be a comment that most PTCs were at stage 1 and of classic and follicular variants, which are less aggressive.
We commented on this in the text.
Reviewer 3 Report
see download file.
to summarize :
the results are really unclearly presented, with many errors, and this makes the paper little interesting. The authors need to change the entire presentation to make the paper more interesting. the discussion will have to be modified according to the new version of the results

Author Response
Reviewer #3
the authors described a multicentric italian study of patients with concomitant MTC and PTC, with demographic, clinical, epidemiological, pathological characteristics and clinical outcomes. Introduction: no comment
Methods: Concerning the methods, the criteria chosen to carry out the study are not clear, and poorly explained, and consequently, the results are little comprehensible.
We have revised the text to enhance clarity.
Results: The main problem concern the tables : poorly understood layout, errors on numbers. Numerous items without interests: all these points must be modified Table 1 : Age of the patients : 39+ 143 =182 (number of patients = 183 !) Same problem with: non-oncologic morbidity : 13+11+5+114+48=191 (number of patients = 183 !) Many items without interests or unclear : non oncologic morbidity, oncologic morbidity, familial thyroid disease, Abbreviations without explanation : FT4, EU, AbTPO, AbTg Table 3 : Error of typology : Hystology, follicolar PTC undefined differentiated : this variant of PTC doesn’t exist RET mutation : 24+88+72= 184 Table 4 : some numbers in the table are not easily comprehensible : death with MTC n=8 (3%) but later in the table death with MTC n=17 (10%). Overall cancer-specific survival outcome: Disease free 109 60, Biochemical disease 32 18, Distant metastasis 10 5, Locoregional disease 8 4 – Death by MTC 6 3, Not evaluable 18 10. PTC outcome: Disease free 143 78, Biochemical disease 17 9, Distant metastasis 1 1, Locoregional disease 1 1, Not evaluable 21 11, MTC outcome: Disease free 119 65, Biochemical disease 24 13, Distant metastasis 10 5, Locoregional disease 7 4 – Death by MTC 17 10, Not evaluable 6 3. Presence of disease by cancer histotype: Disease free 109 60, MTC 42 23, PTC 10 6, Both 4 2, Not evaluable 18
We have largely revised the table according to your suggestions and those of the other reviewers.
Table S1: the germinal and somatic mutations have to be mentioned
Unfortunately, this information was not available due to the retrospective nature of the study. Analysis was performed both on tissues and blood samples.
Figure 1: the three curves can be presented in the same figure.
Done.
the results are really unclearly presented, with many errors, and this makes the paper little interesting. The authors need to change the entire presentation to make the paper more interesting.
We have largely revised the results to address your comments and enhance clarity.
Discussion: the discussion will have to be modified according to the new version of the results
We have revised the Discussion according to the new version of the Results.
Some comments:
« have followed the patients for a long period (median follow-up, 32 months from diagnosis » : 32 months is a very short period for thyroid cancers.
We have revised the text to better clarify our thoughts.
Round 2
Reviewer 1 Report
Authors properly responded to the reviewer's comments.
Reviewer 2 Report
Authors replied to most my comments but not all.
1. My comment about the line 90- Authors write that in MOST cases patients were submitted to total thyroidectomy and central lymph node dissection, however, there is an information that only 33% of patients underwent this procedure. Authors write that the sentence was reformulated, however the meaning of the sentence has not changed.
2. Supplementary table 3- there is no description of table columns. Still there is no information which mutations are somatic and which germline.
3. Supplementary Table 1- the smaller table is non-informative. Melanoma is a tumor site? This should be corrected.
4. Authors give a lot of information about the studied cohort, however, they do not use this information in the discussion. For example the information about RET mutations. Which patients harbored these mutations? Were these RET-positive cases more aggressive, or with familial thyroid diseases??? The discussion should be improved.
5. It is difficult to characterize the cohort of patients with PTC and MTC when you do not compare it to patients with only PTC or with only MTC with the latter comparison being more interesting. If the Authors do not have their own collection of patients with MTC only there is always the possibility to use data from the literature.
Reviewer 3 Report
If many comments have been modified , some persist (not done). The main problem is that tables are always too complicated with many useless characteristics.
Table 1 :
· The term « Diagnosis: » is not adequate : change for circumstances of diagnosis
· "Tyroid goiter" : thyroid instead of tyroid
· Pre-surgical CEA (carcinoembryonic antigen): with 140 « unknown » this item is not interesting
· Presence of non oncologic comorbidity, Presence of oncologic comorbidity, Familiar thyroid cancer FT4 (free thyroxine) etc ...: many of these characteristics are not used in the discussion, they are not useful. Or they have to be discussed.
Table 2. Patient’s previous treatments (N=183) : except « Type of surgery », the other elements are not used in the discussion, they are not useful . Or they have to be discussed.
Table 3 :
PTC undefined differentiated : this variant of PTC doesn’t exist and have to be modified (this comment was not modified). The « Other PTC variants » must be fused with undefined as « PTC (other) ».
What is stage 1, 2, 3, 4 ?
RET mutation : somatic or germinal must be precised
Figure 1. Overall progression-free survival (A) and specific progression-free survival for PCT (B) and 160 MTC only (C) : the three curves must be presented in the same figure (this comment was not modified)
Supplementary Table 2. Data on TNM staging. : it is necessary to add these informations in table 3. Which TNM was used ? 7 or 8th edition ? same question for the stage.
Discussion :
1- the discussion needs to be improved : many results are not analyzed in the discussion (see above). authors need to discuss their results.
2- Moreover some points of the discussion need to be clarified
« Line 203-204 p10 : « However, it should be noted that most PTCs were
204 at stage 1 and of classic and follicular variants, which are less aggressive. » : less aggressive than what ?
Conclusion : « Line 214 Further dedicated studies, e.g. based on miRNA, should investigate the molecular signature of concomitant MTC and PTC » : this sentence is unclear, unrelated to the rest of the text : why only miRNA ?